# An Empirical Study of the Impact of Urbanization on Industry Water Footprint in China

**Daxue Kan [1],*** and **Weichiao Huang [2,3]**

[1]  School of Economics and Trade, Nanchang Institute of Technology, No.289 Tianxiang Road, Nanchang 330099, China

[2]  Department of Economics, Western Michigan University, 1903 W. Michigan Ave., Kalamazoo, MI 49008-5330, USA; huang@wmich.edu

[3]  Specially appointed professor, City College, Wuhan University of Science and Technology, No.947 Heping Avenue, Wuhan 430081, China

*  Correspondence: kdx1983@126.com

**Abstract:** How to advance new urbanization initiatives and reduce the water footprint of industries is one urgent issue about urbanization that needs to be resolved. Based on spatial dynamic panel data, we used the system GMM (Generalized Method of Moments) to study the impact of urbanization on the industrial water footprint. The results show that, overall, urbanization increases the industrial water footprint, industrial virtual water footprint, and industrial gray water footprint in China. There are sectoral and regional differences in the impact of urbanization. Specifically, urbanization reduces the agricultural water footprint and agricultural virtual water footprint but raises the agricultural gray water footprint. Urbanization increases the manufacturing water footprint, manufacturing virtual water footprint, and gray water footprint. Urbanization reduces the virtual water footprint of the service industry but increases the water footprint and gray water footprint in the service industry. At the regional level, urbanization increases the industrial water footprint and gray water footprint across the three major regions. In the eastern region, urbanization has little effect on increasing the industrial water footprint, and reduces the industrial virtual water footprint, whereas in the central and western regions urbanization increases the industrial virtual water footprint. In all three regions, urbanization reduces the agricultural water footprint, increases the manufacturing and service water footprints, reduces the virtual water footprints of agriculture and services, and increases the gray water footprint of agriculture, manufacturing, and services. In the eastern region, the reducing effect of urbanization is the greatest and the increasing effect of urbanization is the smallest. Additionally, in the eastern region, urbanization has reduced the virtual water footprint of manufacturing, whereas in the central and western regions urbanization has increased the virtual water footprint of manufacturing.

**Keywords:** urbanization; industrial water footprint; virtual water footprint; gray water footprint

## 1. Introduction

According to the China Statistical Yearbook, the urbanization rate of China accelerated from less than 18% in 1978 to 59.6% in 2018. At the same time, China's total water consumption increased substantially, from $443.7 \times 10^9$ m$^3$ in 1978 to $601.6 \times 10^9$ m$^3$ in 2018, with per capita water consumption reaching 431.9 m$^3$ in 2018. At present, the amount of water resources that can be developed and utilized in China accounts for less than 40% of the total water resources, and the per capita water resources are only 25% of the world average. In China, only seven provinces (such as Jiangxi and Fujian) do not have water shortage problem, while the other 24 provinces (such as Beijing and Tianjin)

are facing worrisome shortages. In addition to water shortage, China has a rather serious water pollution problem, accentuating the contradiction between water supply and demand in the process of urbanization. Indeed, water shortage and pollution have emerged as major bottlenecks impeding sustainable development and the promotion of a new round of urbanization. In 2018, China's Class IV, Class V, and inferior Class V water bodies that did not meet the drinking water source standards accounted for 25.8%, 33.3%, and 30.1% of rivers, lakes, and provincial waters, respectively. Among the 1935 national surface water monitoring sections, 29% registered water quality readings of Class IV, Class V, and inferior Class V (not meeting the drinking water source standards). Among the 10,168 national groundwater-quality monitoring sites, 86.2% are classified as IV-V. The water quality of 2833 shallow groundwater monitoring wells in China is generally poor, and 76.1% are classified as IV-V. On top of that, acid rain causes further damage to water quality. In 2018, acid rain affected an area totaling 530,000 km$^2$, accounting for 5.5% of the land area (data source: Bulletin on China's Ecological Environment). China's urban water supply is mainly made of surface water or groundwater or a mixture of these two water sources. Water pollution causes further deterioration of the water quality, harming water ecological environments. This is detrimental to implementing a new round of urbanization and water ecological civilization. Despite these problems, China's present urbanization rate is still lagging behind developed countries by nearly 20 percentage points, and it is expected that the urbanization process will continue to advance rapidly. Thus, it is important to understand the linkages between urbanization and water resources and formulate strategies to reconcile the contradiction between water supply and demand and to reduce the water footprint (the water footprint refers to the water resources needed in the production process) in the process of urbanization. To that end, this paper will empirically study the impact of urbanization on the water footprint of industries.

## 2. Literature Review

### 2.1. The Influencing Factors on Water Footprint

There have been several studies concerning the influencing factors on water footprints. Scholars have studied the impact of climate change (Bocchiola et al., 2013) [1], policy change (Fulton et al., 2014) [2], human capital (Ali et al., 2016) [3], gross national income (Miglietta et al., 2017) [4], water harvesting technology (Mohammad et al., 2018) [5], agricultural expansion (Nouri et al., 2019) [6], and trade openness (Mourad et al., 2019) [7] on a country's water footprint. In the context of the Chinese economy, scholars have found that population factors (Wang et al., 2014) [8], economic development levels (Zhao et al., 2014) [9], water-conservation technology (Zhi et al., 2014) [10], international trade (Yang et al., 2015) [11], inward and outward foreign direct investment (Zhang et al., 2015; Kan and Huang, 2019) [12,13], climatic conditions (Yang et al., 2016) [14], consumption levels (Wang et al., 2019) [15], industrial structure (Xie et al., 2019) [16], water-use efficiency (Kan and Lv, 2019) [17], geographical location (Zhang et al., 2019) [18], and shale-gas development (Xu et al., 2019) [19] are important influencing factors on the water footprint. The studies have not explicitly examined urbanization as an influencing factor on the water footprint.

### 2.2. The Impact of Urbanization on Water Use

There have been several studies concerning the impact of urbanization on water quality. Most studies show that urbanization has a negative effect on water quality (Cerqueira et al., 2019; Freeman et al., 2019) [20,21]. Scholars have explored three aspects of the impact of urbanization on water resources utilization. The first aspect is the impact of urbanization on the amount of water resources utilization. Some studies found that urbanization led to the increase of total water use, and the impact was linear (Yang and Ding, 2014; Ma, 2014) [22,23]. However, some other studies found that the impact of urbanization on water resources utilization had a threshold effect, which was non-linear (Kan and Lv, 2017) [24]. In addition, urbanization's impacts vary by different types of water consumption and by different levels of water consumption (Jin et al., 2018; Zhang et al., 2019) [25,26]. The second aspect

is the impact of urbanization on the efficiency of water resources utilization. While some studies found that urbanization improves water-use efficiency (Bao and Chen, 2017; Wang, 2020) [27,28], Ding et al. (2019) found that both population urbanization and land urbanization have a negative impact on industrial water-utilization efficiency [29], some others have found that the relationship between urbanization and water-use efficiency is inverted N-shaped (Cao, 2017) [30]. The third aspect is the impact of urbanization on the structure of water use. Most studies show that urbanization decreases the proportion of agricultural water use and increases the proportion of industrial water use and household water use (Lu et al., 2016; Cao, 2017) [31,32].

The above brief review reveals some gaps in the current literature. First, studies concerning the influencing factors on water footprints have not explicitly examined urbanization as an influencing factor. Second, studies concerning the impact of urbanization on water use have not examined its impact in the perspective of water footprints per se. Although some studies include population as an influencing factor on water footprints, it is only one aspect of urbanization and cannot be equated with the entire extent of urbanization. Three recent papers attempted to address these gaps in the literature. Yu (2014) studied the impact of urbanization on the water footprint in China's Hebei Province [33] and Kan and Lv (2017) analyzed the impact of urbanization on the water footprint and its benefits based on city-level data [34]. However, they only analyzed the issue at the macro level, and they did not explore the issue from an industrial perspective. Our paper intends to enrich the existing literature by doing the following: (1) constructing a spatial dynamic panel model based on provincial industry data from 1997 to 2015, using the system Generalized Method of Moments (GMM) method to examine the overall impact of urbanization on the industrial water footprint in China; (2) further testing the separate impacts of urbanization on the water footprint in the agricultural, manufacturing, and service industries. The objective of this study is to help understand the linkages between urbanization and water resources and formulate countermeasures to reduce the industrial water footprint and to reconcile the contradiction between the water supply and demand in China and its three regions under the process of urbanization, and also provide a reference for other similar countries.

## 3. Study Area

The study covers 31 provinces in China (Figure 1), including 11 provinces (Beijing, Tianjin, Hebei, Liaoning, Shanghai, Jiangsu, Zhejiang, Fujian, Shandong, Guangdong, and Hainan) in the eastern region, 8 provinces (Shanxi, Jilin, Heilongjiang, Anhui, Jiangxi, Henan, Hubei, and Hunan) in the central region, and 12 provinces (Sichuan, Chongqing, Guizhou, Yunnan, Tibet, Shaanxi, Gansu, Qinghai, Ningxia, Xinjiang, Inner Mongolia, and Guangxi) in the western region. In the following we provide a brief description of the urbanization rate and water resources in the study area.

First, it can be seen from Table 1 that China's urbanization rate was 59.6% in 2018, and the urbanization rates in the eastern, central, and western regions were 70.7%, 56.9%, and 52.3%, respectively. The urbanization rates in the central and western regions were lower than the national average. At the provincial level, Shanghai had the highest urbanization rate and Tibet the lowest. The urbanization rate of 13 provinces was higher than the national average. Second, Table 1 shows that China's total water resources were $2746.3 \times 10^9$ m$^3$ in 2018, with per capita water resources reaching 1971.9 m$^3$, which was only 25% of the world average per capita water resources. China's total water consumption was $601.6 \times 10^9$ m$^3$, with per capita water consumption reaching 431.9 m$^3$, of which the total agricultural water consumption was the largest, accounting for 61.4%; total manufacturing water consumption the second largest, accounting for 21.0%; and the total service water consumption was the smallest, accounting for 17.6%. In the three regions, the total water resources, per capita water resources, and per capita water consumption in the eastern region were the smallest, at $517.1 \times 10^9$ m$^3$, 1062.7 m$^3$, and 356.4 m$^3$, respectively. The total water resources, per capita water resources, and per capita water consumption in the central region were the second largest, at $613.9 \times 10^9$ m$^3$, 1543.0 m$^3$, and 472.9 m$^3$, respectively. The total water resources, per capita water resources, and per capita water consumption in the western region were the largest, at $1615.3 \times 10^9$ m$^3$, 3310.1 m$^3$, and 646.3 m$^3$, respectively. The

per capita water resources and water consumption in the eastern region were lower than the national average. The per capita water resources in the central region were lower than the national average, but the per capita water consumption in this region was higher than the national average. The per capita water resources and water consumption in the western region were higher than the national average. In addition, the total water consumption in the eastern region was the largest, at $211.5 \times 10^9$ m$^3$, followed by that in the western region, at $195.8 \times 10^9$ m$^3$, and that in the central region was the smallest, at $194.3 \times 10^9$ m$^3$. Among the regions, the eastern region and the central region had the largest total agricultural water consumption, the second largest total manufacturing water consumption, and the least total service water consumption. The western region had the largest total agricultural water consumption, the second largest total service water consumption, and the least total manufacturing water consumption. At the provincial level, Tibet had the largest total water resources, Ningxia had the least total water resources, and the total water resources of 10 provinces were higher than the national average level. Tibet had the largest per capita water resources, Tianjin had the least per capita water resources, and the total water consumption of 12 provinces was higher than the national average. Xinjiang had the largest per capita water consumption, Beijing had the least per capita water consumption, and the per capita water consumption of 15 provinces was higher than the national average. In terms of sectors, Xinjiang had the largest total agricultural water consumption, Beijing had the least total agricultural water consumption, and the total agricultural water consumption of 14 provinces was higher than the national average level. Jiangsu had the largest total manufacturing water consumption, Tibet had the least total manufacturing water consumption, and the total manufacturing water consumption of 12 provinces was higher than the national average level. Guangdong had the largest total service water consumption, Tibet had the least total service water consumption, and the total service water consumption of 14 provinces was higher than the national average.

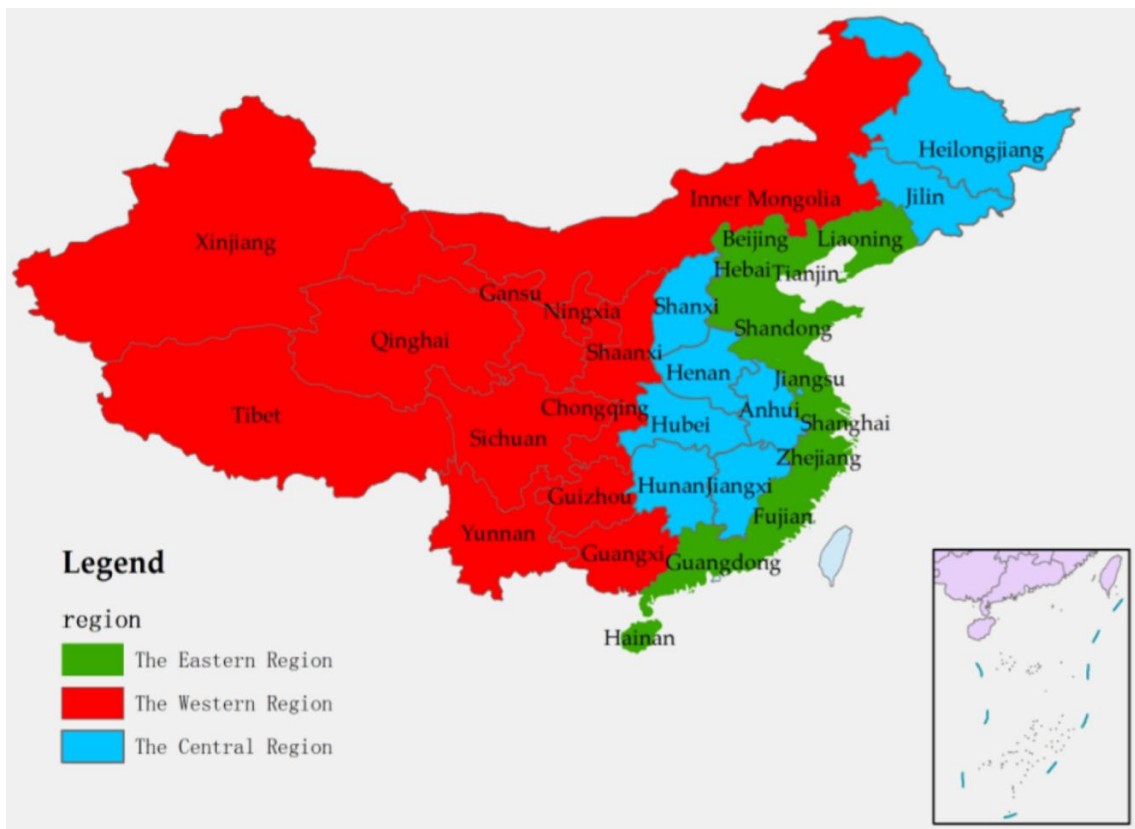

**Figure 1.** Study area.

**Table 1.** Urbanization rate and water resources in China.

| Regions | Urbanization Rate (%) | Total Water Resources (m³) | Total Water Consumption (m³) | Total Agricultural Water Consumption (m³) | Total Manufacturing Water Consumption (m³) | Total Service Water Consumption (m³) | Per Capita Water Resources (m³) | Per Capita Water Consumption (m³) |
|---|---|---|---|---|---|---|---|---|
| Beijing | 86.5 | $3.6 \times 10^9$ | $3.9 \times 10^9$ | $0.4 \times 10^9$ | $0.3 \times 10^9$ | $3.2 \times 10^9$ | 164.2 | 181.8 |
| Tianjing | 83.1 | $1.8 \times 10^9$ | $2.8 \times 10^9$ | $1.0 \times 10^9$ | $0.5 \times 10^9$ | $1.3 \times 10^9$ | 112.9 | 182.2 |
| Hebei | 56.4 | $16.4 \times 10^9$ | $18.2 \times 10^9$ | $12.1 \times 10^9$ | $1.9 \times 10^9$ | $4.2 \times 10^9$ | 217.7 | 242.0 |
| Shanxi | 58.4 | $12.2 \times 10^9$ | $7.4 \times 10^9$ | $4.3 \times 10^9$ | $1.4 \times 10^9$ | $1.7 \times 10^9$ | 328.6 | 200.3 |
| Inner Mongolia | 62.7 | $46.2 \times 10^9$ | $19.2 \times 10^9$ | $14.0 \times 10^9$ | $1.6 \times 10^9$ | $3.6 \times 10^9$ | 1823.0 | 758.8 |
| Liaoning | 68.1 | $23.5 \times 10^9$ | $13.0 \times 10^9$ | $8.1 \times 10^9$ | $1.9 \times 10^9$ | $3.1 \times 10^9$ | 539.4 | 298.6 |
| Jilin | 57.5 | $48.1 \times 10^9$ | $12.0 \times 10^9$ | $8.4 \times 10^9$ | $1.7 \times 10^9$ | $1.8 \times 10^9$ | 1775.3 | 440.9 |
| Heilongjiang | 60.1 | $101.1 \times 10^9$ | $34.4 \times 10^9$ | $30.5 \times 10^9$ | $2.0 \times 10^9$ | $1.9 \times 10^9$ | 2675.1 | 909.6 |
| Shanghai | 88.1 | $3.9 \times 10^9$ | $10.3 \times 10^9$ | $1.7 \times 10^9$ | $6.2 \times 10^9$ | $2.5 \times 10^9$ | 159.9 | 427.1 |
| Jiangsu | 69.6 | $37.8 \times 10^9$ | $59.2 \times 10^9$ | $27.3 \times 10^9$ | $25.5 \times 10^9$ | $6.4 \times 10^9$ | 470.6 | 736.3 |
| Zhejiang | 68.9 | $86.6 \times 10^9$ | $17.4 \times 10^9$ | $7.7 \times 10^9$ | $4.4 \times 10^9$ | $5.3 \times 10^9$ | 1520.5 | 305.1 |
| Anhui | 54.7 | $83.6 \times 10^9$ | $28.6 \times 10^9$ | $15.4 \times 10^9$ | $9.1 \times 10^9$ | $4.1 \times 10^9$ | 1328.9 | 454.4 |
| Fujian | 65.8 | $77.9 \times 10^9$ | $18.7 \times 10^9$ | $8.8 \times 10^9$ | $6.2 \times 10^9$ | $3.7 \times 10^9$ | 1982.9 | 476.1 |
| Jiangxi | 56.0 | $114.9 \times 10^9$ | $25.1 \times 10^9$ | $16.1 \times 10^9$ | $5.9 \times 10^9$ | $3.1 \times 10^9$ | 2479.2 | 541.1 |
| Shandong | 61.2 | $34.3 \times 10^9$ | $21.3 \times 10^9$ | $13.4 \times 10^9$ | $3.3 \times 10^9$ | $4.7 \times 10^9$ | 342.4 | 212.1 |
| Henan | 51.7 | $34.0 \times 10^9$ | $23.5 \times 10^9$ | $12.0 \times 10^9$ | $5.0 \times 10^9$ | $6.4 \times 10^9$ | 354.6 | 244.8 |
| Hubei | 60.3 | $85.7 \times 10^9$ | $29.7 \times 10^9$ | $15.4 \times 10^9$ | $8.7 \times 10^9$ | $5.6 \times 10^9$ | 1450.2 | 502.4 |
| Hunan | 56.0 | $134.3 \times 10^9$ | $33.7 \times 10^9$ | $19.5 \times 10^9$ | $9.3 \times 10^9$ | $4.9 \times 10^9$ | 1952.0 | 489.9 |
| Guangdong | 70.7 | $189.5 \times 10^9$ | $42.1 \times 10^9$ | $21.4 \times 10^9$ | $9.9 \times 10^9$ | $10.7 \times 10^9$ | 1683.4 | 373.9 |
| Guangxi | 50.2 | $183.1 \times 10^9$ | $28.8 \times 10^9$ | $19.6 \times 10^9$ | $4.8 \times 10^9$ | $4.4 \times 10^9$ | 3732.6 | 586.7 |
| Hainan | 59.1 | $41.8 \times 10^9$ | $4.5 \times 10^9$ | $3.3 \times 10^9$ | $0.3 \times 10^9$ | $1.0 \times 10^9$ | 4495.7 | 485.0 |
| Chongqing | 65.5 | $52.4 \times 10^9$ | $7.7 \times 10^9$ | $2.5 \times 10^9$ | $2.9 \times 10^9$ | $2.3 \times 10^9$ | 1697.2 | 250.0 |
| Sichuan | 52.3 | $295.3 \times 10^9$ | $25.9 \times 10^9$ | $15.7 \times 10^9$ | $4.3 \times 10^9$ | $6.0 \times 10^9$ | 3548.2 | 311.4 |
| Guizhou | 47.5 | $97.9 \times 10^9$ | $10.7 \times 10^9$ | $6.1 \times 10^9$ | $2.5 \times 10^9$ | $2.0 \times 10^9$ | 2726.2 | 297.5 |
| Yunnan | 47.8 | $220.7 \times 10^9$ | $15.6 \times 10^9$ | $10.7 \times 10^9$ | $2.1 \times 10^9$ | $2.8 \times 10^9$ | 4582.3 | 323.4 |
| Tibet | 31.1 | $465.8 \times 10^9$ | $3.2 \times 10^9$ | $2.7 \times 10^9$ | $0.2 \times 10^9$ | $0.3 \times 10^9$ | 136804.7 | 931.0 |
| Shaanxi | 58.1 | $37.1 \times 10^9$ | $9.4 \times 10^9$ | $5.7 \times 10^9$ | $1.5 \times 10^9$ | $2.2 \times 10^9$ | 964.8 | 243.4 |
| Gansu | 47.7 | $33.3 \times 10^9$ | $11.2 \times 10^9$ | $8.9 \times 10^9$ | $0.9 \times 10^9$ | $1.4 \times 10^9$ | 1266.6 | 426.8 |
| Qinghai | 54.4 | $96.2 \times 10^9$ | $2.6 \times 10^9$ | $1.9 \times 10^9$ | $0.3 \times 10^9$ | $0.4 \times 10^9$ | 16018.3 | 434.6 |
| Ningxia | 58.9 | $1.5 \times 10^9$ | $6.6 \times 10^9$ | $5.7 \times 10^9$ | $0.4 \times 10^9$ | $0.5 \times 10^9$ | 214.6 | 966.4 |
| Xinjiang | 50.9 | $85.9 \times 10^9$ | $54.9 \times 10^9$ | $49.1 \times 10^9$ | $1.3 \times 10^9$ | $4.5 \times 10^9$ | 3482.6 | 2225.5 |
| The Eastern Region | 70.7 | $517.1 \times 10^9$ | $211.5 \times 10^9$ | $105.1 \times 10^9$ | $60.4 \times 10^9$ | $46.1 \times 10^9$ | 1062.7 | 356.4 |
| The Central Region | 56.9 | $613.9 \times 10^9$ | $194.3 \times 10^9$ | $121.5 \times 10^9$ | $43.1 \times 10^9$ | $29.6 \times 10^9$ | 1543.0 | 472.9 |
| The Western Region | 52.3 | $1615.3 \times 10^9$ | $195.8 \times 10^9$ | $142.7 \times 10^9$ | $22.6 \times 10^9$ | $30.4 \times 10^9$ | 3310.1 | 646.3 |
| China | 59.6 | $2746.3 \times 10^9$ | $601.6 \times 10^9$ | $369.3 \times 10^9$ | $126.2 \times 10^9$ | $106.1 \times 10^9$ | 1971.9 | 431.9 |

Data source: Authors' calculations according to the data of the China Statistical Yearbook.

## 4. Methods

### 4.1. Model Construction

Based on the study of Kan and Lv (2018) [35] and the generalized spatial panel specification (Lesage and Pace, 2009) [36], we constructed the following model specifying the industrial water footprint (*WF*) as the dependent variable and urbanization (*UR*) as the focus independent variable:

$$WF_{ijt} = C + \gamma \times WF_{ijt-1} + \rho W \times WF_{ijt} + \beta_1 \times UR_{it} + \lambda \times X_t + \mu_i + \delta_j + \varphi_t + \varepsilon_{it} \tag{1}$$

$$\varepsilon_{it} = \varphi W \varepsilon_{it} + v_{it} \tag{2}$$

When $\rho \neq 0$, $\beta_1 \neq 0$, $\varphi = 0$ and when $\rho = 0$, $\beta_1 \neq 0$, $\varphi \neq 0$, the above model is converted respectively into a spatial dynamic panel lag model and a spatial dynamic panel error model, the former indicating that a province's industrial water footprint was not only related to that province's urbanization but also related to the water footprints of adjacent provinces, and the latter indicating that a province's industrial water footprint was not only related to that province's urbanization but was also related to the water footprints and urbanization of adjacent provinces.

In the above model, *i* represents the province, *j* represents the industry, *t* represents the year, and *X* contains two categories of control variables, one category being the provincial level control variables, including water endowment (*WB*), resident income level (*PI*), and climate factors (*QH*) and the other category being industry-level control variables, including the industry's size (*ES*), the industry's structure (*IS*), industrial water efficiency (*YF*), industrial technology advancement (*TE*), industrial environmental regulation (*EI*), industrial import and export trade (*TR*), and the foreign investment of industry (*FO*). $\mu$, $\delta$, and $\varphi$ are province, industry, and time dummy variables, respectively. $\varepsilon$ and *W* are random disturbance terms and the spatial weight matrix, respectively. Considering the possible lingering effect of changes in the industrial water footprint, we added its lag term to the model. The addition of a lag term also enabled the incorporation of other influencing factors that were not explicitly included in the model. In addition, considering the possible heteroscedasticity of variables, provincial-level control variables, such as water endowment and resident income levels, and industry-level control variables, such as industry size and industry structure, were presented in logarithmic form in the model.

### 4.2. Variable Measurement and Data Collection

The first dependent variable, the industrial water footprint, was measured as: industrial internal water footprint + industrial external water footprint + industrial internal gray water footprint + industrial external gray water footprint. The row vector of each industry's water consumption and the row vector of wastewater discharge in the traditional input-output model were added to measure the industrial water footprint. To measure the internal water footprint of the industry (the amount of water needed for the final unit production of industry *j*), we first calculated the direct consumption coefficient matrix (the direct consumption coefficient was the input required by industry *j* from industry *d* to increase the unit output) and used the matrix to obtain the Leontief inverse matrix, then multiplied the Leontief inverse matrix with the direct virtual water intensity matrix to obtain the virtual water intensity of industry *j* (the direct virtual water intensity was the amount of water input directly needed by industry *j* to increase the unit output. The virtual water intensity was the sum of all direct and indirect water quantities required to meet the final demand of industry *j*'s unit output). The internal water footprint was obtained by multiplying the virtual water intensity of industry *j* by the domestic consumer demand. To measure the external water footprint of the industry (the amount of water needed for imported products used in the final demand of industry *j*), we added the amount of water needed for the imported products directly used for the final demand of industry *j* with the amount of water needed for the imported products used as the intermediate demand and to be converted into the final consumption of industry *j* (as imported products used for intermediate demand are converted

into final demand and exports, they were adjusted by (domestic consumer demand-export)/domestic consumption demand). To measure the industry's internal gray water footprint (the amount of wastewater discharged from domestically produced products used in the final demand of industry *j*), we first obtained the virtual wastewater intensity of industry *j* by multiplying the Leontief inverse matrix with the direct virtual wastewater intensity matrix (direct virtual wastewater intensity was the amount of wastewater directly discharged by industry *j* from increasing the unit output; the virtual wastewater intensity was the sum of all direct and indirect wastewater volumes that met the final demand of the industry *j* unit output); then multiplied the virtual wastewater intensity of industry *j* by the domestic consumption demand (the amount of wastewater discharged from imported products used in the final demand of industry *j*) to obtain the industry's internal gray water footprint. As for measuring the industry's external wastewater footprint, we added the amount of wastewater discharged by the imported products directly used in the final demand of industry *j* with the amount of wastewater discharged by the imported products that were used for the intermediate demand and then converted this into the final consumption of industry *j* (as imported products used for intermediate demand are converted into final demand and exports, they were adjusted by (domestic consumer demand-export)/domestic consumption demand).

The second dependent variable, the virtual water footprint of an industry, was measured as: virtual water export-virtual water import. The former was the amount of water needed for industry *j*'s export products, which was equal to the virtual water intensity of industry *j* times the export demand. The latter was the external water footprint of the industry. In addition to analyzing the impact of urbanization on the industrial water footprint overall, we further investigated the separate impact of urbanization on the agricultural, manufacturing, and service industries. Using the same measurement algorithms as above, we obtained the water footprints of the agricultural, manufacturing, and service industries, the virtual water footprints and the gray water footprints of these three industries. The original data came from China's Input-Output Table and its provincial input-output tables, the China Statistical Yearbook and its provincial statistical yearbooks, the China Water Resources Bulletin and its provincial water resources bulletins, China's Statistical Bulletin of Water Conservancy Development, annual reports of provincial water conservancy statistics, the China Environmental Yearbook, and the China Environmental Statistics Yearbook. The sample period for this study was 1997-2015, and the input-output tables provided data for 1997, 2000, 2002, 2005, 2007, 2010, 2012, and 2015, respectively. The missing data in the intervening years were estimated using the moving average method.

Turning to measuring urbanization—the focus independent variable—most of the existing literature has measured it by the urbanization rate of the resident population. This paper draws on the index system constructed by Lv and Kan (2017) [37] and adds into the three-level index system of population urbanization—economic urbanization, social urbanization, and spatial urbanization—with the following variables: the urbanization rate of household registration, the proportion of high-tech industry value added in above-scale manufacturing industries and the comprehensive coverage rate of social insurance, the number of telephones per 100 households (including mobile phones), and the rate of environmental noise reaching and beyond the standard level. We used a principal component approach to process the data and the reverse index by the Z-score method and constructed the "1-reverse index" or the "1/reverse index" of urbanization. The original data were derived from the China Statistical Yearbook and its provincial statistical yearbooks and the China Economic and Social Development Statistics Database.

Finally, regarding the measurement of control variables, the water resources endowment, income level of residents, and climate factors were measured by, respectively, per capita water resources, per capita disposable income of urban residents + per capita net income of rural areas (through population weighting processing), and precipitation. The industry's scale, industry's structure, industry's water-use efficiency, and industry's technology progress (industry value added/annual average number of employees in the industry) were measured by, respectively, the industry's total output value, the industry's output value/total output value, the industry's value added/industry's

water consumption, and the total labor productivity. The industry's environmental regulation, import and export trade, and foreign capital utilization level were measured by, respectively, the industry's pollution control investment/total amount of industry pollution, the industry's import and export volume, and the output of foreign capital unit/industry's total output value. The original data were derived from the China Statistical Yearbook and the provincial statistical yearbooks, the China Industrial Statistical Yearbook, the China Environmental Yearbook, the China Environmental Statistics Annual Report, the CEIC (China entrepreneur Investment Club) China Economic Database (this study covers 30 provinces; Tibet was excluded due to incomplete data. In the sample some variables, such as industry scales and import and export totals had zero values. As $\ln(1+T) \approx T$, when T was very small, the industry scales and total imports and exports with zero values were given 1 before taking the logarithmic transformation).

### 4.3. Spatial Autocorrelation Test

The improvement of the urbanization level in a region not only comes from the supply of local factors and water resources, but also depends on the supply of other regions' factors and water resources. The complementary or competitive relationship between regions leads to commodity circulation, factor mobility, and water resources flow, which have an important impact on the development of regional urbanization. Due to similar social, economic, and geographical conditions, the urbanization development goals and water-resource management goals set by a region are usually based on the urbanization development level and water-resource management level of the surrounding regions, and the policies to promote urbanization development and water-resource management are often learned from each other between geographically adjacent regions. Therefore, urbanization and the water footprint are likely to have spatial correlation. In the spatial-econometrics literature, Moran's I index is commonly used to test the existence of a spatial correlation of regional economic variables. Following standard practice, we used Moran's I index to study the spatial autocorrelation pattern between urbanization and the industrial water footprint, industrial virtual water footprint, and industrial gray water footprint (when calculating Moran's I index, the spatial weight matrix $W$ adopted the 0-1 weight matrix commonly used in the literature—$W = 1$ if two provinces are adjacent and $W = 0$ otherwise. When Moran's I index was >0, <0, and = 0 respectively, it shows that provincial variables had spatial positive correlation, negative correlation, and non-correlation). The results showed that the Moran's I value of urbanization and the industrial water footprint, industrial virtual water footprint, and industrial gray water footprint were all positive in the sample period, indicating that there were spatial clusters between urbanization and the industrial water footprint, industrial virtual water footprint, and industrial gray water footprint in China. That is, there was statistically significant spatial interdependence between urbanization and the industrial water footprint, industrial virtual water footprint, and industrial gray water footprint across provinces. Specifically, in the provinces with higher urbanization levels, the adjacent provinces also had a higher level of urbanization (for example, Beijing had a higher level of urbanization, and the adjacent province Tianjin also had a higher level of urbanization; Shanghai had a higher level of urbanization, and the adjacent provinces of Jiangsu and Zhejiang also had higher levels of urbanization) and vice versa (for example, Yunnan had a lower level of urbanization, and the adjacent provinces of Guizhou and Guangxi also had a lower level of urbanization; Gansu had a lower level of urbanization, and the adjacent provinces of Qinghai and Xinjiang also had a lower level of urbanization). Similarly, spatial interdependence also applied to the industrial water footprint, industrial virtual water footprint, and industrial gray water footprint variables. That is, in the provinces with higher industrial water footprints, industrial virtual water footprints, and industrial gray water footprints, the industrial water footprints, industrial virtual water footprints, and industrial gray water footprints in adjacent provinces were also higher (for example, the industrial water footprint in Jiangsu was higher, and the industry water footprint in the adjacent province of Shandong was also higher) and vice versa (for example, the industrial water footprint in Shaanxi was lower, and the industrial water footprints in the adjacent provinces of Shanxi and

Ningxia were also lower). Further, provinces with higher urbanization levels appeared to have spatial correlation with provinces with higher industrial water footprints, industrial virtual water footprints, and industrial gray water footprints. Likewise, provinces with a lower urbanization level also had spatial correlation with provinces with lower industrial water footprints, industrial virtual water footprints, and industrial gray water footprints. Therefore, simple autocorrelation tests have already provided preliminary evidence that urbanization is positively correlated with the industrial water footprint, industrial virtual water footprint, and industrial gray water footprint.

### 4.4. Selection of Spatial Dynamic Panel Model

Before using the system GMM regression, we conducted LM (Lagrange Multiplier) tests to determine whether the spatial dynamic panel lag model or spatial dynamic panel error model was a more appropriate estimation model. It can be seen from Table 2 that when the dependent variables were the industrial water footprint and the agricultural, manufacturing, and service industries' water footprints, the significant levels of LM (lag) and Robust LM (lag) were higher than LM (error) and Robust LM (error), respectively. Further, in the case that the dependent variables were the industrial virtual water footprint; agricultural, manufacturing, and service virtual water footprints; the industrial gray water footprint; and the agricultural, manufacturing, and service gray water footprints, only the LM (lag) statistic was significant. Thus, the spatial dynamic panel lag model appeared more appropriate and was selected for estimation. Researchers commonly used the GMM method to estimate the spatial dynamic panel lag model. The GMM method can be divided into the differential GMM method and the system GMM (Arellano and Bond, 1991; Arellano and Bover, 1995; Blundell and Bond, 1998) [38–40]. The estimator of the system GMM method further uses the moment condition of the level equation on the basis of the estimator of the differential GMM method and takes the first-order difference of the lagged variable as the instrumental variable for the corresponding level variable in the level equation. Therefore, the system GMM method was used here to estimate the model. The results are shown in Table 3.

**Table 2.** LM (Lagrange Multiplier) statistics for model selection.

|  | LM (lag) | LM (error) | Robust LM (lag) | Robust LM (error) |
|---|---|---|---|---|
| Industrial water footprint | 12.424 *** | 7.098 ** | 6.453 ** | 3.564 * |
| Agricultural water footprint | 12.003 *** | 6.834 ** | 6.276 ** | 3.451 * |
| Manufacturing water footprint | 10.215 *** | 6.012 ** | 5.802 ** | 2.792 * |
| Service water footprint | 8.796 *** | 5.181 ** | 5.005 ** | 2.413 * |
| Industrial virtual water footprint | 6.138 * | 2.347 | —— | —— |
| Agricultural virtual water footprint | 4.550 * | 1.369 | —— | —— |
| Manufacturing virtual water footprint | 5.199 * | 1.975 | —— | —— |
| Service virtual water footprint | 6.632 * | 2.494 | —— | —— |
| Industrial gray water footprint | 5.337 * | 2.027 | —— | —— |
| Agricultural gray water footprint | 4.363 * | 1.335 | —— | —— |
| Manufacturing gray water footprint | 4.984 * | 1.913 | —— | —— |
| Service gray water footprint | 6.341 * | 2.410 | —— | —— |

Note: *, **, and *** indicate that the variable was significant at the level of 10%, 5%, and 1%, respectively. Data source: Authors' collation according to the software regression results.

**Table 3.** Estimation results of urbanization impact at national and regional levels.

| | China | | | The Eastern Region | | | The Central Region | | | The Western Region | | |
|---|---|---|---|---|---|---|---|---|---|---|---|---|
| | Industrial Water Footprint | Industrial Virtual Water Footprint | Industrial Gray Water Footprint | Industrial Water Footprint | Industrial Virtual Water Footprint | Industrial Gray Water Footprint | Industrial Water Footprint | Industrial Virtual Water Footprint | Industrial Gray Water Footprint | Industrial Water Footprint | Industrial Virtual Water Footprint | Industrial Gray Water Footprint |
| C | 2.646 ** | 3.021 ** | 3.271 * | 3.119 ** | 4.568 ** | 2.523 * | 2.884 ** | 3.073 * | 2.977 * | 4.349 ** | 3.056 ** | 3.485 ** |
| Dependent variable with one lag period | 0.281 * | 0.266 ** | 0.252 ** | 0.304 * | 0.272 * | 0.289 ** | 0.275 * | 0.262 ** | 0.311 ** | 0.280 * | 0.257 ** | 0.243 * |
| ln*UR* | 0.198 * | 0.104 ** | 0.137 ** | 0.103 ** | −0.076 ** | 0.065 * | 0.201 ** | 0.138 ** | 0.142 ** | 0.257 ** | 0.184 * | 0.196 * |
| ln*WB* | 0.112 * | 0.093 | 0.106 | 0.075 ** | 0.092 | 0.064 | 0.137 | 0.119 | 0.125 | 0.141 ** | 0.128 | 0.139 |
| ln*PI* | 0.107 ** | 0.105 * | 0.098 * | −0.081 * | −0.073 * | −0.076 ** | 0.122 * | 0.116 * | 0.108 * | 0.130 * | 0.119 * | 0.113 ** |
| ln*QH* | 0.063 | 0.067 * | 0.062 | 0.045 | 0.037 | 0.039 | 0.074 | 0.071 | 0.075 | 0.089 ** | 0.077 * | 0.082 |
| ln*ES* | 0.169 ** | 0.174 ** | 0.145 ** | 0.102 ** | 0.106 ** | 0.090 ** | 0.188 ** | 0.195 ** | 0.167 ** | 0.196 * | 0.208 ** | 0.181 ** |
| ln*IS* | 0.096 ** | 0.095 ** | 0.090 ** | −0.047 ** | −0.042 ** | −0.051 * | 0.123 ** | 0.120 ** | 0.126 * | 0.131 ** | 0.133 ** | 0.138 * |
| ln*YF* | −0.134 * | −0.136 * | −0.131 * | −0.180 * | −0.178 * | −0.193 ** | −0.125 ** | −0.127 * | −0.119 ** | −0.097 ** | −0.085 * | −0.082 * |
| ln*TE* | −0.093 | −0.091 | −0.084 | −0.165 ** | −0.159 ** | −0.172 ** | −0.086 | −0.078 | −0.082 | −0.061 | −0.056 | −0.054 |
| ln*EI* | −0.121 | −0.098 | −0.129 | −0.146 ** | −0.132 * | −0.164 * | −0.109 | −0.085 | −0.114 | −0.093 | −0.082 | −0.097 |
| ln*TR* | 0.130 ** | 0.125 ** | 0.118 ** | 0.074 | 0.046 | 0.063 | 0.141 ** | 0.134 ** | 0.129 ** | 0.160 ** | 0.157 ** | 0.145 ** |
| ln*FO* | 0.077 | 0.078 | 0.075 * | 0.039 | 0.030 | 0.026 | 0.089 | 0.092 | 0.083 * | 0.084 | 0.086 | 0.089 |
| $\rho$ | 0.065 ** | 0.067 ** | 0.070 * | 0.078 ** | 0.094 ** | 0.080 ** | 0.084 ** | 0.086 ** | 0.085 * | 0.118 ** | 0.069 ** | 0.076 ** |
| Wald test | 1332.824 | 1024.571 | 1008.253 | 1047.664 | 1156.966 | 1262.204 | 970.297 | 954.838 | 992.161 | 1095.670 | 1148.602 | 882.967 |
| Hansen test | 0.717 | 0.579 | 0.608 | 0.681 | 0.619 | 0.702 | 0.571 | 0.599 | 0.668 | 0.609 | 0.634 | 0.515 |

Note: *, **, and *** indicate that the variable was significant at the level of 10%, 5%, and 1%, respectively. Arellano-Bond AR Statistics are not abnormal. Data source: Authors' collation of the data according to the software regression results.

## 5. Results

*5.1. The Impact of Urbanization on the Industrial Water Footprint*

(1) Empirical results at the national level: Table 3 shows that as the urbanization level increased by 1%, the industrial water footprint, industrial virtual water footprint, and industrial gray water footprint increased by 0.198%, 0.104%, and 0.137%, respectively, with the significance levels at 10%, 5%, and 5%, respectively. It appears that urbanization raised the industrial water footprint, industrial virtual water footprint, and industrial gray water footprint.

(2) Empirical results at the regional level: As can be seen from Table 3, in the eastern region, as the level of urbanization increased by 1%, the industrial water footprint, industrial virtual water footprint, and industrial gray water footprint changed by 0.103%, −0.076%, and 0.065%, respectively. All of the changes were statistically significant. In the central region, a 1% increase in urbanization caused the industrial water footprint, industrial virtual water footprint, and industrial gray water footprint to significantly increase by 0.201%, 0.138%, and 0.142%, respectively. In the western region, a 1% increase in urbanization caused the industrial water footprint, industrial virtual water footprint, and industrial gray water footprint to significantly increase by 0.257%, 0.184%, and 0.196%, respectively. The results show that urbanization in the three regions raised the industrial water footprint and gray water footprint, while the improvement effect of urbanization was smaller in the eastern region. The results also show that urbanization in the eastern region contributed to the reduction of the virtual water footprint of industry, while urbanization in the central and western regions increased industry's virtual water footprint.

(3) Empirical results on spatial spillover effects: Table 3 shows that the parameter ρ was positively significant in all equations, indicating that a given province's industrial water footprint, industrial virtual water footprint, and industrial gray water footprint were respectively affected by the industrial water footprint, industrial virtual water footprint, and industrial gray water footprint of the adjacent provinces. This means that the industrial water footprint, industrial virtual water footprint, and industrial gray water footprint of the provinces were significantly affected by the industrial water footprint, industrial virtual water footprint, and industrial gray water footprint of the adjacent provinces, respectively. That is, there were significant spatial spillover effects from adjacent provinces in industrial water footprint, industrial virtual water footprint, and industrial gray water footprint.

*5.2. The Impact of Urbanization on the Water Footprint in Different Industries*

We grouped all industries into agricultural, manufacturing, and service industries according to the "Three Industries Classification Guidelines" stipulated by the National Bureau of Statistics of China. We then examined the separate impacts of urbanization on the water footprint, the virtual water footprint, and the gray water footprint of the agricultural, manufacturing, and service industries. The results are shown in Tables 4–6.

**Table 4.** Estimation results of the impact of urbanization on the water footprints in agriculture, manufacturing, and services.

| | China | | | The Eastern Region | | | The Central Region | | | The Western Region | | |
|---|---|---|---|---|---|---|---|---|---|---|---|---|
| | Agricultural Water Footprint | Manufacturing Water Footprint | Services Water Footprint | Agricultural Water Footprint | Manufacturing Water Footprint | Services Water Footprint | Agricultural Water Footprint | Manufacturing Water Footprint | Services Water Footprint | Agricultural Water Footprint | Manufacturing Water Footprint | Services Water Footprint |
| C | 2.508 ** | 2.861 * | 3.049 ** | 2.954 ** | 4.320 ** | 2.397 * | 2.732 * | 2.910 ** | 2.819 * | 4.113 ** | 2.891 ** | 3.298 * |
| Dependent variable with one lag period | 0.277 * | 0.263 ** | 0.255 * | 0.296 ** | 0.268 * | 0.282 ** | 0.271 ** | 0.254 * | 0.305 ** | 0.279 * | 0.256 * | 0.244 ** |
| ln$UR$ | −0.092 ** | 0.158 ** | 0.136 ** | −0.115 * | 0.113 ** | 0.104 ** | −0.070 * | 0.159 ** | 0.131 ** | −0.052 ** | 0.164 ** | 0.140 * |
| Control variable | Yes | Yes | Yes | Yes | Yes | Yes | Yes | Yes | Yes | Yes | Yes | Yes |
| $\rho$ | 0.074 * | 0.085 * | 0.084 ** | 0.093 ** | 0.112 * | 0.093 * | 0.105 ** | 0.103 ** | 0.102 * | 0.130 * | 0.082 ** | 0.079 ** |
| Wald test | 1359.489 | 1045.082 | 1028.430 | 1068.611 | 1180.117 | 1287.461 | 989.716 | 973.947 | 1012.016 | 1117.596 | 1171.586 | 900.638 |
| Hansen test | 0.743 | 0.603 | 0.632 | 0.706 | 0.644 | 0.728 | 0.595 | 0.623 | 0.693 | 0.634 | 0.659 | 0.537 |

Note: *, **, and *** indicate that the variable was significant at the level of 10%, 5%, and 1%, respectively. Arellano-Bond AR Statistics are not abnormal. Data source: Authors' collation according to the software regression results.

**Table 5.** Estimation results of the impact of urbanization on the virtual water footprint in agriculture, manufacturing, and services.

| | China | | | The Eastern Region | | | The Central Region | | | The Western Region | | |
|---|---|---|---|---|---|---|---|---|---|---|---|---|
| | Agricultural Virtual Water Footprint | Manufacturing Virtual Water Footprint | Services Virtual Water Footprint | Agricultural Virtual Water Footprint | Manufacturing Virtual Water Footprint | Services Virtual Water Footprint | Agricultural Virtual Water Footprint | Manufacturing Virtual Water Footprint | Services Virtual Water Footprint | Agricultural Virtual Water Footprint | Manufacturing Virtual Water Footprint | Services Virtual Water Footprint |
| C | 2.470 ** | 2.817 ** | 3.003 * | 2.909 * | 4.252 ** | 2.361 * | 2.690 * | 2.866 ** | 2.777 * | 4.049 * | 2.850 ** | 3.248 * |
| Dependent variable with one lag period | 0.277 * | 0.263 * | 0.251 ** | 0.294 * | 0.269 * | 0.285 ** | 0.272 * | 0.254 * | 0.305 ** | 0.267 * | 0.256 * | 0.245 * |
| ln$UR$ | −0.035 ** | 0.209 * | −0.072 ** | −0.057 ** | −0.043 ** | −0.096 * | −0.031 ** | 0.228 ** | −0.063 ** | −0.024 ** | 0.251 * | −0.047 ** |
| Control variable | Yes | Yes | Yes | Yes | Yes | Yes | Yes | Yes | Yes | Yes | Yes | Yes |
| $\rho$ | 0.073 * | 0.076 ** | 0.085 * | 0.081 ** | 0.108 * | 0.091 ** | 0.095 ** | 0.100 ** | 0.099 * | 0.128 ** | 0.082 ** | 0.086 ** |
| Wald test | 1235.542 | 949.800 | 934.667 | 971.202 | 1072.525 | 1170.080 | 899.483 | 885.159 | 919.750 | 1015.703 | 1064.774 | 818.523 |
| Hansen test | 0.686 | 0.554 | 0.581 | 0.648 | 0.591 | 0.668 | 0.547 | 0.572 | 0.636 | 0.582 | 0.605 | 0.495 |

Note: *, **, and *** indicate that the variable was significant at the level of 10%, 5%, and 1%, respectively. Arellano-Bond AR Statistics are not abnormal. Data source: Authors' collation according to the software regression results.

**Table 6.** Estimation results of the impact of urbanization on the gray water footprints in agriculture, manufacturing, and services.

| | China | | | The Eastern Region | | | The Central Region | | | The Western Region | | |
|---|---|---|---|---|---|---|---|---|---|---|---|---|
| | Agricultural Gray Water Footprint | Manufacturing Gray Water Footprint | Services Gray Water Footprint | Agricultural Gray Water Footprint | Manufacturing Gray Water Footprint | Services Gray Water Footprint | Agricultural Gray Water Footprint | Manufacturing Gray Water Footprint | Services Gray Water Footprint | Agricultural Gray Water Footprint | Manufacturing Gray Water Footprint | Services Gray Water Footprint |
| C | 2.598 * | 2.957 * | 3.151 ** | 3.052 * | 4.460 * | 2.476 ** | 2.823 * | 3.007 * | 2.915 ** | 4.249 * | 2.994 * | 3.403 ** |
| Dependent variable with one lag period | 0.290 ** | 0.276 * | 0.262 * | 0.313 ** | 0.282 * | 0.298 * | 0.284 * | 0.272 ** | 0.319 * | 0.291 ** | 0.268 * | 0.256 * |
| ln$UR$ | 0.054 ** | 0.192 ** | 0.168 ** | 0.036 * | 0.085 ** | 0.073 ** | 0.057 ** | 0.206 ** | 0.174 ** | 0.075 * | 0.240 ** | 0.218 ** |
| Control variable | Yes | Yes | Yes | Yes | Yes | Yes | Yes | Yes | Yes | Yes | Yes | Yes |
| $\rho$ | 0.075 * | 0.083 ** | 0.069 * | 0.081 ** | 0.117 ** | 0.095 ** | 0.101 ** | 0.104 * | 0.103 * | 0.132 ** | 0.093 ** | 0.090 * |
| Wald test | 1296.856 | 996.931 | 981.027 | 1019.394 | 1125.753 | 1228.142 | 944.117 | 929.075 | 965.390 | 1066.104 | 1117.606 | 859.173 |
| Hansen test | 0.715 | 0.580 | 0.606 | 0.679 | 0.618 | 0.701 | 0.572 | 0.604 | 0.666 | 0.610 | 0.635 | 0.519 |

Note: *, **, and *** indicate that the variable was significant at 10%, 5%, and 1% level, respectively. Arellano-Bond AR Statistics are not abnormal. Data source: Authors' collation according to the software regression results.

Table 4 presents the empirical results of the impact of urbanization on the water footprint of the agricultural, manufacturing, and service industries separately. It shows that as urbanization increased by 1%, the water footprint of the agricultural, manufacturing, and service industries increased by −0.092%, 0.158%, and 0.136%, respectively. All of these coefficients were statistically significant, indicating that urbanization reduced the agricultural water footprint and increased the manufacturing and service water footprints in China. The separate impacts of urbanization on the water footprints of these three types of industries at the national level were also found to be similar at the regional level. One notable regional difference was that urbanization had the largest effect on reducing the agricultural water footprint, and the smallest effect on increasing the manufacturing and service water footprints in the eastern region.

Table 5 presents the results of the impact of urbanization on the virtual water footprint of the agricultural, manufacturing, and service industries. It shows that as urbanization increased by 1%, the virtual water footprints of the agricultural, manufacturing, and service industries changed by −0.035%, 0.209%, and −0.072%, respectively. All of these coefficients were statistically significant, indicating that urbanization had reduced the virtual water footprints of the agricultural and service industries, but increased the manufacturing virtual water footprint in China. This national pattern was also applicable to the central and western regions. Table 5 also shows that as urbanization increased by 1% in the eastern region, the virtual water footprint of the agricultural, manufacturing, and service industries in that region decreased by 0.057%, 0.043%, and 0.096%, respectively. All of these coefficients were statistically significant, indicating that urbanization had reduced the virtual water footprint of the agricultural, manufacturing, and service industries in the eastern region.

Table 6 displays the results of the impact of urbanization on the gray water footprints of the agricultural, manufacturing, and service industries. It shows that as urbanization increased by 1%, the gray water footprints of the agricultural, manufacturing, and service industries increased by 0.054%, 0.192%, and 0.168%, respectively. All of these coefficients were positively significant, indicating that urbanization had raised the gray water footprints of the agricultural, manufacturing, and service industries in China. This national pattern was also similarly exhibited in three regions, among which urbanization in the eastern region had the smallest impact on the gray water footprints of the agricultural, manufacturing, and service industries.

## 6. Discussion

Table 3 shows that urbanization increased the industry water footprint, industry virtual water footprint, and industry gray water footprint in China. This may be because while urbanization increases the efficiency of water resources utilization in industry by promoting industrial agglomeration, generating economies of scale, reducing the cost of technological research and development, improving the technological level of industry, gaining more opportunities for education and training, promoting the upgrading of industrial structures, and reducing the misplacement of water resources in industry caused by factor market distortions, etc., urbanization also promotes the use of water resources in industries by stimulating consumption, expanding the scale of industries, promoting the transfer of employment to labor-intensive industries and traditional services that tend to consume more water, stimulating fixed asset formation in industries, and promoting foreign capital inflow and export scale expansion. The former has a smaller improvement effect, while the latter exerts a larger effect, resulting in a net increase in the water, virtual water, and gray water footprints. The underlying reason is that China's urbanization is more of an extensive mode of development, lacking sufficient connotation and sophistication. China's urbanization focuses on the urbanization growth rate per se, resulting in low quality urbanization. The average quality of urbanization in the sample period was only 1.263, and the quality of urbanization varied greatly across regions. In addition, the demographic dividend and market formed by urbanization have led to an influx of foreign investment in labor-intensive industries and a large expansion in the export of low value-added products, resulting in a large amount of water consumption in the production process of foreign enterprises in these industries and in the

production process of export products, which in turn has resulted in a large-scale expansion of virtual water exports. In the process of urbanization, China imported a large number of high value-added products with a high technology content. Most of these industries are modern manufacturing and technology-intensive industries, which consume less water. As a result, the scale of virtual water import is small, causing urbanization to improve the virtual water footprint of industry. Finally, the urbanization process in China is characterized by weak industrial water ecological awareness, imperfect sewage treatment facilities, a lack of industrial water saving and reclaimed water reuse facilities, a low utilization rate (according to the China Statistical Yearbook, China's reclaimed water reuse rate only accounted for about 11% of the sewage treatment capacity in 2015), low implementation efficiency of industrial environmental regulations, and arbitrary enforcement of the law on water pollution. All these have led to a notable rise in the industrial gray water footprint.

Urbanization raised the industrial water footprint and gray water footprint in all three regions, while the eastern region showed the smallest increase in the water footprint, perhaps because the quality of urbanization in this region was higher than that in the central and western regions (the average quality of urbanization during the sample period was 1.697 in the eastern region, the average quality of urbanization during the sample period in the central and western regions was 1.112 and 0.975, respectively). The eastern region achieved the largest improvement in the utilization efficiency of water resources in industry through an industrial agglomeration effect, industrial technology-level upgrading, industrial human capital enhancing, industrial structure upgrading, and the reduction of industrial water resources misplacement. Additionally, the eastern region achieved better containment of the industrial gray water footprint through the improvement of industrial sewage treatment facilities, better facilities for water saving and reclaimed water reuse, higher utilization rate (according to the China Statistical Yearbook, the reclaimed water reuse rate of the eastern region accounted for about 25% of the sewage treatment capacity in 2015), stricter industrial environmental regulations, a higher implementation efficiency, and a sound water pollution supervision system.

Urbanization in the eastern region reduced the industrial virtual water footprint, and urbanization in the central and western regions raised the industrial virtual water footprint. This may be because, on the one hand, urbanization in the eastern region attracted high-quality types of foreign investment (such as asset-seeking and efficiency-seeking foreign investment), and these types of foreign investment are centered on technology/knowledge-intensive industries producing and exporting technology/knowledge-intensive products. Foreign enterprises in these industries consume less water resources in the production process of export products, making the virtual water export smaller. On the other hand, rapid urbanization in the eastern region quickly dissipated the population dividend. Coupled with the rising cost of land and raw materials, this pushed water-consuming labor-intensive industries out of the region to the central and western regions and Southeast Asian countries. Again, this relocation led to the decline of the virtual water footprint in the eastern region.

Table 4 shows that urbanization reduced the agricultural water footprint and increased the water footprints of manufacturing and service industries in China as a whole and in all three regions. Additionally, urbanization had the largest effect on reducing the agricultural water footprint, and the smallest effect on increasing the manufacturing water footprint and service water footprint in the eastern region. Compared with the central and western regions, the eastern region has absorbed a large surplus rural labor force in the urbanization process and raised the level of agricultural mechanization and production efficiency (according to the China Statistical Yearbook, the average agricultural production efficiency during the sample period was $920.9 per capita in the eastern region; the agricultural production efficiency during the sample period in the central and western regions was $731.0 per capita and $791.8 per capita, respectively), thereby substantially reducing the agricultural water footprint. Moreover, fast economic growth enabled the eastern region to amass substantial fiscal revenue, which was used to finance construction of agricultural water conservancy infrastructure and water-saving irrigation infrastructure, raised the effective utilization rate of irrigation water, and significantly reduced the agricultural water footprint in this region. In addition, compared with the

central and western regions, urbanization in the eastern region led to a greater replacement of farmland by urban construction and a greater decrease in arable land. Studies found that as urbanization in the eastern region increased by 1%, arable land decreased by nearly 140,000 hectares. Arable land in the central and western regions did not decrease as much. This decrease of arable land area caused by urbanization in the eastern region also contributed to the greater reduction of the agricultural water footprint in this region. Furthermore, urbanization in the eastern region increased people's income and upgraded residents' consumption structure toward a healthier consumption pattern. This, in turn, increased the demand for low calorie, fat and sugar, and less water-consuming agricultural products such as rice, legumes, potatoes, barley, broad beans, and wheat and reduced demand for more water-consuming meat products (according to the Food and Agriculture Organization of the United Nations, 10,000–15,000 kg of water is needed to produce 1 kg of meat (its effective utilization rate is less than 0.01%), while only 400–3000 kg of water is needed to produce 1 kg of grain product, which is about 5% of the water needed to produce meat).

The eastern region, compared to the central and western regions, experienced the smallest increase in manufacturing and service water footprints. While the proportion of manufacturing industry in the eastern region initially increased with urbanization, subsequently it declined, and the proportion of service industries rose rapidly. Additionally, the proportion of modern high-technology manufacturing industry among manufacturing industries in the eastern region was higher, and the proportion of knowledge-intensive emerging service industry among service industries was larger. These industries tend to consume less water than other manufacturing and service industries. Moreover, the eastern region has a better developed ladder water pricing mechanism for the manufacturing and service industries and stricter regulations on high-water-consuming low-end manufacturing and traditional service industries, enhancing the utilization efficiency of water-saving facilities and the recycling efficiency of water resources. All these enabled the region to achieve the smallest rise in the water footprints of the manufacturing and service industries.

Table 5 shows that urbanization reduced the virtual water footprints of the agricultural and service industries and raised the virtual water footprint of the manufacturing industry in China at the country level and also in the central and western regions. This could be due to the fact that agricultural exports were less than imports, and this trade deficit increased along with the process of urbanization. In 2015, China exported $70.68 \times 10^9$ and imported $116.88 \times 10^9$, with a deficit of $46.2 \times 10^9$ agricultural products (data source: China Customs). Major agricultural export items (such as vegetables, fruits, aquatic products, and tea) are products that consume less water, and major import items (such as cereals, livestock products, cotton, sugar, edible oilseeds, and edible vegetable oil) are products that consume more water. Consequently, the agricultural virtual water footprint has decreased. Similarly, as urbanization progresses, the service trade deficit increases steadily. According to the China Service Trade Statistics 2016, in 2015, China's service exports were $288.19 \times 10^9$ and imports were $424.81 \times 10^9$, amounting to a deficit of $136.62 \times 10^9$. Among them, exports of less water-consuming services (such as telecommunications, computer and information services, professional management, and consulting services) were greater than imports (with a surplus of US$29.72 \times 10^9$), and exports of water-consuming services (such as traditional tourism and transportation services) were much smaller than imports (with a trade deficit of $74.94 \times 10^9$). Consequently, the service virtual water footprint fell. As for urbanization increasing the manufacturing virtual water footprint, this could be attributed to the manufacturing trade structure and trade surplus. In 2015, China's manufacturing exports were $2.209 \times 10^{12}$ and imports were $1.563 \times 10^{12}$, registering a surplus of $0.6459 \times 10^{12}$ (data source: Compiled from the 2016 China Statistical Yearbook). The main export items (such as textile, clothing and footwear, automatic data processing equipment and its components, and hand-held or vehicle-mounted radios) are traditional labor-intensive manufacturing products, which consume more water per unit, while the main import items (such as integrated circuits, crude oil, iron ore, and coal) are high-technology-content and energy products, which consume less water per unit. Consequently, the manufacturing virtual water footprint increased.

The notable exception of urbanization actually reducing the manufacturing virtual water footprint in the eastern region may be due to the fact that urbanization in this region has optimized the internal structure of the manufacturing industry by forming an industrial agglomeration, upgrading the technological level, and promoting the industrial gradient transfer. This has transformed the industry from labor-intensive low-end manufacturing to high-end manufacturing with a high technology content. Although the eastern region also has a manufacturing trade surplus, the virtual water content of manufacturing exports is lower than that of manufacturing imports because less water-consuming middle- and high-end manufacturing products are exported, and more water-consuming processing trade has been transferred to the central and western regions.

Table 6 shows that urbanization in China raised the gray water footprints of the agricultural, manufacturing, and service industries in China in all three regions, with the eastern region sustaining the smallest impact on the gray water footprints in the agricultural, manufacturing, and service industries. During the process of urbanization, the extensive production mode of agriculture has not been fundamentally changed. Agricultural non-point source pollution (from planting, livestock and poultry breeding, aquaculture, straw, plastic film, and many other aspects) has resulted in serious agricultural water pollution, and yet the water pollution monitoring system has not been established in rural areas. All of this made the agricultural gray water footprint rise significantly. The eastern region achieved the lowest rise in the agricultural gray water footprint due to its more advanced development in agricultural eco-recycling and stricter management and control of the use of pesticides and chemical fertilizers. Urbanization has led to the rapid expansion of China's manufacturing industries (particularly industries such as papermaking, chemical industry, iron and steel, electric power, food, textiles, and other industries), which have generated enormous wastewater in the production process, resulting in a substantial increase in the manufacturing gray water footprint. However, compared with the central and western regions, the eastern region has invested more in the wastewater treatment of the manufacturing industry. In 2015, the eastern region invested $1081.96 \times 10^6$ in wastewater treatment, which amounted to an average investment of $98.36 \times 10^6$ per province in the region. The central and western regions invested $346.35 \times 10^6$ and $472.89 \times 10^6$, respectively. The average investment per province in the central and western regions was $57.72 \times 10^6$ and $33.78 \times 10^6$, respectively, which were only 58.69% and 34.34% of the counterpart in the eastern region (data source: Compiled from the 2016 China Statistical Yearbook). In addition, cities in the eastern region have established a more comprehensive water pollution control and supervision system, with stricter environmental controls and a higher utilization rate of manufacturing sewage treatment facilities, enabling the region to achieve the smallest increase in the manufacturing gray water footprint. Finally, while urbanization promotes rapid development of China's modern service industry, it also expands traditional service industries (such as lodging and catering, wholesale and retail, transportation, and post and telecommunications), resulting in a great deal of water pollution and raising the gray water footprint of the service industry. Compared to the central and western regions, the eastern region has implemented a step pricing policy for water use to control the total amount of water used. At the same time, this region has acquired and developed more advanced water pollution control equipment and imposed stricter regulations on the water ecological environment. All these enabled this region to achieve the smallest increase in gray water footprint of service industry.

## 7. Conclusions

How to advance the new urbanization initiative and reduce the water footprints of industries are urgent urbanization issues that need to be resolved. Based on spatial dynamic panel data, we used the system GMM method to study the impact of urbanization on the industrial water footprint. The results show that, overall, urbanization increases the industrial water footprint, industrial virtual water footprint, and industrial gray water footprint in China. In separate industries: urbanization reduces the agricultural water footprint and agricultural virtual water footprint but increases the agricultural gray water footprint; urbanization increases the manufacturing water footprint, manufacturing virtual water

footprint, and manufacturing gray water footprint; urbanization increases the water footprint and gray water footprint of the service industry, but reduces the virtual water footprint of the service industry in China. At the regional level, urbanization increases the industrial water footprint and industrial gray water footprint in all three regions. In the eastern region urbanization exerts the least effect on raising the industrial water footprint and actually reduces the industrial virtual water footprint. Urbanization in the central and western regions increases the industrial virtual water footprint. In all three regions urbanization has reduced the agricultural water footprint and raised the manufacturing water footprint and service water footprint. The eastern region saw the greatest effect of urbanization on reducing the agricultural water footprint and the smallest increase in the manufacturing water footprint and service water footprint. Additionally, in the eastern region urbanization has reduced the virtual water footprints of agriculture, manufacturing, and services. In the central and western regions, urbanization has reduced the virtual water footprints of agriculture and services but increased the manufacturing virtual water footprint. Finally, in all three regions urbanization has increased the gray water footprints of agriculture, manufacturing, and services, with the eastern region sustaining the least of such an effect.

Based on these findings, some policy implications and prescriptions are as follows. First, to reduce the overall industrial water footprint while promoting new urbanization initiatives, China, especially the central and western regions, should strive to reduce the cost of industry agglomeration, build a high-end industry agglomeration platform, implement innovation-driven development strategy, increase investment in industry R&D, and optimize the direction and structure of investment. To reduce the industrial water footprint, China should also promote the market-oriented reform of water resources management and raise the utilization efficiency of water resources through upgrading the industrial structure and technology level, the enhancement of human capital, and the curtailing of water resources misplacement.

Second, to reduce the agricultural water footprint while promoting new urbanization, China, especially the central and western regions, needs to raise the level of agricultural mechanization, develop and improve the construction of agricultural water conservancy and water-saving irrigation infrastructure, and raise the effective utilization rate of farmland irrigation water. In addition, as people's income increases it is necessary to guide the populace to optimize consumption structure and consumption patterns toward less water-consuming agricultural products and away from more water-consuming meat products.

Third, to reduce the water footprint of the manufacturing and service industries while promoting new urbanization, China, especially the central and western regions, needs to optimize the internal structure of the manufacturing and service industries, increase the proportion of less water-consuming modern manufacturing and service industries, implement the water ladder pricing mechanism, strictly monitor and contain the water consumption of the low-end manufacturing industry and traditional service industry, and improve the utilization rate of water-saving facilities and recycling efficiency of water resources in the manufacturing and service industries.

Fourth, to reduce the virtual water footprint of industry in the process of urbanization, China, especially the central and western regions, needs to optimize the export structure of agricultural, manufacturing, and services products by reducing the export of water-consuming agricultural products, increasing the export of less water-consuming agricultural products, by reducing the export of water-consuming labor-intensive manufacturing products, and increasing the export of less water-consuming technology-intensive manufacturing products, and also by reducing the export of traditional service industries that consume more water and increasing the export of new service industries that consume less water.

Fifth, to reduce the gray water footprint of industry, China, especially the central and Western regions, needs to raise awareness of the water ecological environment in the process of urbanization, further improve the sewage treatment pricing system and sewage treatment facilities, improve the facility utilization rate, and strengthen environmental regulations and enforcement. Specifically, in

the agricultural sector, authorities need to direct farmers to change extensive modes of production; improve the monitoring system of agricultural water pollution; control the use of pesticides and chemical fertilizers with a move toward precise fertilization; and strive for technological advancement in several areas including green prevention and control technology, livestock, and poultry manure treatment technology, aquaculture tail water treatment technology, and straw comprehensive utilization technology, etc. In the manufacturing and service sectors, they need to increase investment in wastewater treatment, improve water pollution treatment facilities, and enhance the omnibus management and supervision system of water pollution so as to reduce the gray water footprint in the manufacturing and service industries.

With the development of new urbanization, urbanization has changed from an extensive mode to an intensive mode. The world is facing many challenges from global climate change: global warming will directly affect the quantity of water resources, lead to a reduction of glaciers and decrease the water flow of rivers and lakes, and cause drought; it will also cause more water evaporation and decrease precipitation; it will also cause floods in coastal areas and pollute fresh water resources. We hope the above proposed policy measures will help to reduce the industrial water footprint and to reconcile the contradiction between water supply and demand in the process of urbanization.

**Author Contributions:** D.K. analyzed the data and drafted the manuscript; W.H. gave many valuable comments on the draft and also polished it. All authors have read and agreed to the published version of the manuscript.

**Funding:** The National Natural Science Foundation of China (No.71764018), the Social Science Foundation of Jiangxi Province (No.18JL03), the National Statistical Science Foundation of China (No.2018LZ01), the Foundation of Jiangxi Provincial Soft Science Research Base of Water Safety and Sustainable Development (No.19JDYB03), the Science and Technology Foundation of Jiangxi Education Department (No.GJJ171001), and the Humanities and Social Sciences Foundation of Jiangxi Province (No. JJ17119).

**Acknowledgments:** The authors greatly appreciate the funding support. The authors would also like to extend special thanks to the editor and anonymous reviewers for their constructive comments and suggestions for improving the quality of this paper.

**Conflicts of Interest:** The authors declare no conflicts of interest.

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
