# Peer review of "An Empirical Study of the Impact of Urbanization on Industry Water Footprint in China"

_sustainability, doi:10.3390/su12062263_

Round 1
Reviewer 1 Report
It is an important and interesting topic since soon the majority of the world will be urbanized and they need more water. However, the manuscript suffers many problems need to be taken care of before it can be considered for publication in the journal of Sustainability.
Comments
- The manuscript lacks a clear objective which makes it difficult to follow the organization and the findings of the study. The statements outlined in lines105-109 are more of a methodology more than an objective.
- The abstract is very long and it is confusing for the reader to understand. This because of repeating the findings. I suggest it should be short, concise, precise and to the point.
- The introduction section has included important background information, but all of them have not been backed up by references.
- The international conceptual framework (section 2.2 should come before 2.1-China experience- and it did not respond directly to the main theme of the manuscript which stated by the title of the manuscript).
- The authors did not tell us whether they borrow this model (lines 112 -114) from other studies or is it of their own?
- This study covers 30 provinces. Therefore, it is important for the reader to understand the geographical location, therefore, addition of a section of study area which shows some details such as a map showing the different provinces, the amounts and sources of water supply, the number of urban population and the type of industry in these provinces. Also, the authors may need to have another map that shows the three industrial regions (Line 368)
- The information included in lines 232 – 241 are very general. In order to be a meaningful and more scientific, they need to be backed up by examples from the findings of the study.
- The authors have used the Moran’ I index. They did not justify the rationale of using this model (i.e. what are the results of the past studies locally or internationally that have applied this model).
- The authors forgot to write the source(s) of all of the tables.
Finally, in one sentence or two, the reader would like to know based on the future challenges of increased urbanization and climate change, what is the prospect for future sustainability of water use for industry and urbanization
Author Response
Response to Reviewer 1 Comments
- The manuscript lacks a clear objective which makes it difficult to follow the organization and the findings of the study. The statements outlined in lines105-109 are more of a methodology more than an objective.
Response 1: In response to the reviewer’s comment, we have stated the objectives explicitly in lines 115-118. Specifically, we have added the following:The objective of this study is to help understand the linkages between urbanization and water resources and formulate countermeasures to reduce industry water footprint and to reconcile the contradiction between water supply and demand in China and its three regions under the process of urbanization, and also provide reference for other similar countries.
- The abstract is very long and it is confusing for the reader to understand. This because of repeating the findings. I suggest it should be short, concise, precise and to the point.
Response 2: In response to the reviewer’s comment, we have cut down and revised the abstract in lines 27-33 as follows: In all three regions, urbanization has reduced agricultural water footprint, increased manufacturing and service water footprint, reduced the virtual water footprint of agriculture and services and increased the gray water footprint of agriculture, manufacturing and services. In the eastern region, the reducing effect of urbanization is the greatest and the increasing effect of urbanization is the least. Also in the eastern region, urbanization has reduced the virtual water footprint of manufacturing, whereas in the central and western regions urbanization has increased the virtual water footprint of manufacturing.
- The introduction section has included important background information, but all of them have not been backed up by references.
Response 3: As suggested by the reviewer, we revised the introduction and other sections in lines 37, 56, 447, 461-462, 482, 520 and 534, adding sources of background information, such as according to China Statistical Yearbook, data source: Bulletin on China’s Ecological Environment, data source: Compiled from the 2016 China Statistical Yearbook.
- The international conceptual framework (section 2.2 should come before 2.1-China experience- and it did not respond directly to the main theme of the manuscript which stated by the title of the manuscript).
Response 4: As suggested by the reviewer, we moved Section 2.2 to the front of Section 2.1 in lines 67-100.
- The authors did not tell us whether they borrow this model (lines 112 -114) from other studies or is it of their own?
Response 5: As suggested by the reviewer, we explicitly mentioned that we construct the model based on the study of Kan and Lv (2018) and the generalized spatial panel specification (Lesage and Pace, 2009) in lines 170-171.
- This study covers 30 provinces. Therefore, it is important for the reader to understand the geographical location, therefore, addition of a section of study area which shows some details such as a map showing the different provinces, the amounts and sources of water supply, the number of urban population and the type of industry in these provinces. Also, the authors may need to have another map that shows the three industrial regions (Line 368).
Response 6: As suggested by the reviewer, we have added a section of study area in lines119-167, Details regarding the urbanization rate, total water resources, total water consumption, per capita water resources, per capita water consumption, total agricultural, manufacturing and service water consumption of all provinces cannot be displayed in two maps as suggested by the reviewer, rather, at least eight maps are needed. In order to economize the space and still provide detailed information of the study area, we add Section 3 and Table 1. The contents are as follows:
First, it can be seen from Table 1 that China's urbanization rate is 59.6% in 2018, and the urbanization rates in the eastern, central and western regions are 70.7%, 56.9% and 52.3% respectively. The urbanization rates in the central and western regions are lower than the national average. At the provincial level, Shanghai has the highest urbanization rate, and Tibet the lowest. The urbanization rate of 13 provinces is higher than the national average.
Second, Table 1 shows that China's total water resources are 2746.3´109 m3 in 2018, with per capita water resources reaching 1971.9 m3, which are only 25% of the world average per capita water resources. China's total water consumption is 601.6´109 m3, with per capita water consumption reaching 431.9 m3, of which total agricultural water consumption is the largest, accounting for 61.4%; total manufacturing water consumption the second, accounting for 21.0%; total service water consumption is the least, accounting for 17.6%. In the three regions, the total water resources, per capita water resources and per capita water consumption in the eastern region are the least, 517.1´109 m3, 1062.7 m3 and 356.4 m3, respectively. The total water resources, per capita water resources and per capita water consumption in the central region are the second largest, 613.9´109 m3, 1543.0 m3 and 472.9 m3, respectively. The total water resources, per capita water resources and per capita water consumption in the western region are the largest, 1615.3´109 m3, 3310.1 m3 and 646.3 m3, respectively. The per capita water resources and water consumption in the eastern region are lower than the national average. The per capita water resources in the central region are lower than the national average, but the per capita water consumption in this region is higher than the national average. The per capita water resources and water consumption in the western region are higher than the national average. In addition, the total water consumption in the eastern region is the largest, 211.5´109 m3, followed by that in the western region, 195.8´109 m3, and that in the central region is the least, 194.3´109 m3. Among the regions, the eastern region and the central region have the largest total agricultural water consumption, the second largest total manufacturing water consumption, and the least total service water consumption. The western region has the largest total agricultural water consumption, the second largest total service water consumption, and the least total manufacturing water consumption. At the provincial level, Tibet has the largest total water resources, Ningxia has the least total water resources, and the total water resources of 10 provinces are higher than the national average level. Tibet has the largest per capita water resources, Tianjin has the least per capita water resources, and the total water consumption of 12 provinces is higher than the national average. Xinjiang has the largest per capita water consumption, Beijing has the least per capita water consumption, and the per capita water consumption of 15 provinces is higher than the national average. In terms of sectors, Xinjiang has the largest total agricultural water consumption, Beijing has the least total agricultural water consumption, and the total agricultural water consumption of 14 provinces is higher than the national average level. Jiangsu has the largest total manufacturing water consumption, Tibet has the least total manufacturing water consumption, and the total manufacturing water consumption of 12 provinces is higher than the national average level. Guangdong has the largest total service water consumption, Tibet has the least total service water consumption, and the total service water consumption of 14 provinces is higher than the national average.
Table 1. Urbanization rate and water resources in China.
|
Regions |
Urbanization Rate (%) |
Total Water Resources (m3) |
Total Water Consumption (m3) |
Total Agricultural Water Consumption (m3) |
Total Manufacturing Water Consumption (m3) |
Total service Water Consumption (m3) |
Per Capita Water Resources (m3) |
Per Capita Water Consumption (m3) |
|
Beijing |
86.5 |
3.6´109 |
3.9´109 |
0.4´109 |
0.3´109 |
3.2´109 |
164.2 |
181.8 |
|
Tianjing |
83.1 |
1.8´109 |
2.8´109 |
1.0´109 |
0.5´109 |
1.3´109 |
112.9 |
182.2 |
|
Hebei |
56.4 |
16.4´109 |
18.2´109 |
12.1´109 |
1.9´109 |
4.2´109 |
217.7 |
242.0 |
|
Shanxi |
58.4 |
12.2´109 |
7.4´109 |
4.3´109 |
1.4´109 |
1.7´109 |
328.6 |
200.3 |
|
Inner Mongolia |
62.7 |
46.2´109 |
19.2´109 |
14.0´109 |
1.6´109 |
3.6´109 |
1823.0 |
758.8 |
|
Liaoning |
68.1 |
23.5´109 |
13.0´109 |
8.1´109 |
1.9´109 |
3.1´109 |
539.4 |
298.6 |
|
Jilin |
57.5 |
48.1´109 |
12.0´109 |
8.4´109 |
1.7´109 |
1.8´109 |
1775.3 |
440.9 |
|
Heilongjiang |
60.1 |
101.1´109 |
34.4´109 |
30.5´109 |
2.0´109 |
1.9´109 |
2675.1 |
909.6 |
|
Shanghai |
88.1 |
3.9´109 |
10.3´109 |
1.7´109 |
6.2´109 |
2.5´109 |
159.9 |
427.1 |
|
Jiangsu |
69.6 |
37.8´109 |
59.2´109 |
27.3´109 |
25.5´109 |
6.4´109 |
470.6 |
736.3 |
|
Zhejiang |
68.9 |
86.6´109 |
17.4´109 |
7.7´109 |
4.4´109 |
5.3´109 |
1520.5 |
305.1 |
|
Anhui |
54.7 |
83.6´109 |
28.6´109 |
15.4´109 |
9.1´109 |
4.1´109 |
1328.9 |
454.4 |
|
Fujian |
65.8 |
77.9´109 |
18.7´109 |
8.8´109 |
6.2´109 |
3.7´109 |
1982.9 |
476.1 |
|
Jiangxi |
56.0 |
114.9´109 |
25.1´109 |
16.1´109 |
5.9´109 |
3.1´109 |
2479.2 |
541.1 |
|
Shandong |
61.2 |
34.3´109 |
21.3´109 |
13.4´109 |
3.3´109 |
4.7´109 |
342.4 |
212.1 |
|
Henan |
51.7 |
34.0´109 |
23.5´109 |
12.0´109 |
5.0´109 |
6.4´109 |
354.6 |
244.8 |
|
Hubei |
60.3 |
85.7´109 |
29.7´109 |
15.4´109 |
8.7´109 |
5.6´109 |
1450.2 |
502.4 |
|
Hunan |
56.0 |
134.3´109 |
33.7´109 |
19.5´109 |
9.3´109 |
4.9´109 |
1952.0 |
489.9 |
|
Guangdong |
70.7 |
189.5´109 |
42.1´109 |
21.4´109 |
9.9´109 |
10.7´109 |
1683.4 |
373.9 |
|
Guangxi |
50.2 |
183.1´109 |
28.8´109 |
19.6´109 |
4.8´109 |
4.4´109 |
3732.6 |
586.7 |
|
Hainan |
59.1 |
41.8´109 |
4.5´109 |
3.3´109 |
0.3´109 |
1.0´109 |
4495.7 |
485.0 |
|
Chongqing |
65.5 |
52.4´109 |
7.7´109 |
2.5´109 |
2.9´109 |
2.3´109 |
1697.2 |
250.0 |
|
Sichuan |
52.3 |
295.3´109 |
25.9´109 |
15.7´109 |
4.3´109 |
6.0´109 |
3548.2 |
311.4 |
|
Guizhou |
47.5 |
97.9´109 |
10.7´109 |
6.1´109 |
2.5´109 |
2.0´109 |
2726.2 |
297.5 |
|
Yunnan |
47.8 |
220.7´109 |
15.6´109 |
10.7´109 |
2.1´109 |
2.8´109 |
4582.3 |
323.4 |
|
Tibet |
31.1 |
465.8´109 |
3.2´109 |
2.7´109 |
0.2´109 |
0.3´109 |
136804.7 |
931.0 |
|
Shaanxi |
58.1 |
37.1´109 |
9.4´109 |
5.7´109 |
1.5´109 |
2.2´109 |
964.8 |
243.4 |
|
Gansu |
47.7 |
33.3´109 |
11.2´109 |
8.9´109 |
0.9´109 |
1.4´109 |
1266.6 |
426.8 |
|
Qinghai |
54.4 |
96.2´109 |
2.6´109 |
1.9´109 |
0.3´109 |
0.4´109 |
16018.3 |
434.6 |
|
Ningxia |
58.9 |
1.5´109 |
6.6´109 |
5.7´109 |
0.4´109 |
0.5´109 |
214.6 |
966.4 |
|
Xinjiang |
50.9 |
85.9´109 |
54.9´109 |
49.1´109 |
1.3´109 |
4.5´109 |
3482.6 |
2225.5 |
|
The Eastern Region |
70.7 |
517.1´109 |
211.5´109 |
105.1´109 |
60.4´109 |
46.1´109 |
1062.7 |
356.4 |
|
The Central Region |
56.9 |
613.9´109 |
194.3´109 |
121.5´109 |
43.1´109 |
29.6´109 |
1543.0 |
472.9 |
|
The Western Region |
52.3 |
1615.3´109 |
195.8´109 |
142.7´109 |
22.6´109 |
30.4´109 |
3310.1 |
646.3 |
|
China |
59.6 |
2746.3´109 |
601.6´109 |
369.3´109 |
126.2´109 |
106.1´109 |
1971.9 |
431.9 |
Data source: Authors’ calculation according to the data of China Statistical Yearbook.
- The information included in lines 232 – 241 are very general. In order to be a meaningful and more scientific, they need to be backed up by examples from the findings of the study.
Response 7: As suggested by the reviewer, we gave specific examples in lines 299-312 as follows: For example, Beijing has a higher level of urbanization, the adjacent province Tianjin also has a higher level of urbanization; Shanghai has a higher level of urbanization, the adjacent provinces Jiangsu and Zhejiang also have a higher level of urbanization. For example, Yunnan has a lower level of urbanization, the adjacent provinces Guizhou and Guangxi also have a lower level of urbanization; Gansu has a lower level of urbanization, the adjacent provinces Qinghai and Xinjiang also have a lower level of urbanization. For example, the industry water footprint in Jiangsu is higher, the industry water footprint in adjacent province Shandong is also higher. For example, the industry water footprint in Shaanxi is lower, the industry water footprint in adjacent provinces Shanxi and Ningxia are also lower.
- The authors have used the Moran’ I index. They did not justify the rationale of using this model (i.e. what are the results of the past studies locally or internationally that have applied this model).
Response 8: In response to the reviewer’s comment, we provided the rationale for using the Moran' I index in lines 276-289 as follows: The improvement of urbanization level in a region not only comes from the supply of local factors and water resources, but also depends on the supply of other regions’ factors and water resources. The complementary or competitive relationship between regions leads to commodity circulation, factors mobility and water resources flow, which have an important impact on the development of regional urbanization. Due to the similar social, economic and geographical conditions, the urbanization development goals and water resource management goals set by a region are usually based on the urbanization development level and water resource management level of the surrounding regions, and the policies to promote urbanization development and water resource management are often learned from each other between geographically adjacent regions. Therefore, urbanization and water footprint are likely to have spatial correlation. In spatial econometrics literature, the Moran’ I index is commonly used to test the existence of spatial correlation of regional economic variables. Following the standard practice, we use the Moran’ I index to study the spatial autocorrelation pattern between urbanization and industry water footprint, industry virtual water footprint and industry grey water footprint.
- The authors forgot to write the source(s) of all of the tables.
Response 9: As suggested by the reviewer, we have added the source below each table. See all tables for details. For example, data source: Authors’ calculation according to the data of China Statistical Yearbook, data source: Authors’ collation according to the software regression results.
- Finally, in one sentence or two, the reader would like to know based on the future challenges of increased urbanization and climate change, what is the prospect for future sustainability of water use for industry and urbanization.
Response 10: As suggested by the reviewer, we added a paragraph in lines 648-654 as follows: With the development of new urbanization, urbanization has changed from extensive mode to intensive mode. The world is facing many challenges from global climate change: global warming will directly affect the quantity of water resources, lead to reduction of glaciers and decrease water flow of rivers and lakes, and cause drought; it will also cause more water evaporation and decrease precipitation; it will also cause floods in coastal areas, and pollute fresh water resources. We hope the above proposed policy measures will help to reduce industry water footprint and to reconcile the contradiction between water supply and demand in the process of urbanization.
Reviewer 2 Report
Dear colleagues,
I really must commend your conclusions related to agriculture on page 12/15 line 524-530 and on the next page 13/15 line 549-555. My congratulations. A good work(!).
Author Response
Response to Reviewer 2 Comments
- I really must commend your conclusions related to agriculture on page 12/15 line 524-530 and on the next page 13/15 line 549-555. My congratulations. A good work(!).
Response 1: Thanks for your comments.
Reviewer 3 Report
The topic of the paper is interesting and it is a good candidate for the publication.
However, they are some issues to be dealt with:
Please eliminate those multiple references. After that, please check the manuscript thoroughly and eliminate ALL the lumps in the manuscript. This should be done by characterising each reference individually. This can be done by mentioning 1 or 2 phrases per reference to show how it is different from the others and why it deserves mentioning. Multiple references are of no use for a reader and can substitute even a kind of plagiarism, as sometimes authors are using them without proper studies of all references used. In the case, each reference should be justified by it is used and at least a short assessment provided.
Remove "billion" as it is often commuted with "milliard" due to discrepancies between the UK and US English usage, use mathematical symbols instead
Authors are using water footprint without proper referencing of the origin, definitions, and specification,
Please be more specific:
"Based on the existing literature and the generalized spatial panel specification, we construct the following model specifying industry water footprint (WF) as the dependent variable and
urbanization (UR) as the focus independent variable:"
In the conclusions, in addition to summarising the actions taken and results, please strengthen the explanation of their significance. It is recommended to use quantitative reasoning comparing with appropriate benchmarks, especially those stemming from previous work.
The paper is focusing on China, however, it should not mean that the majority of references are from that region.
In English text, equations, and pictures, please use British standards for numbers in the text and pictures with delimiters:
1,000,000 rather than 1000000 or 1 000 000 or 1.000.000
For the text clarity would you refrain from using additional words, mostly meaningless filler words, which can be omitted or some archaic words see e.g. “respectively”, “thus”, “hence”, therefore”, “furthermore”, “thereby”, “basically,”, “meanwhile”,” wherein”, “herein”, “hitherto”, “Nonetheless”, “Perceivably”, “whereas”,etc. ?
210.38million RMB Should be 210.38 MRMB, however, you should also provide some indicative exchange rate between RMB and say EUR or USD
statements as
"In 2015, China exported $70.68 billion and imported $116.88 billion, with a
432 deficit of $46.2 billion agricultural products." are presented without providing the reference for the source of the information.
When the indicated issues are properly treated the manuscript can be considered for the publication.
Author Response
Response to Reviewer 3 Comments
- Please eliminate those multiple references. After that, please check the manuscript thoroughly and eliminate ALL the lumps in the manuscript. This should be done by characterising each reference individually. This can be done by mentioning 1 or 2 phrases per reference to show how it is different from the others and why it deserves mentioning. Multiple references are of no use for a reader and can substitute even a kind of plagiarism, as sometimes authors are using them without proper studies of all references used. In the case, each reference should be justified by it is used and at least a short assessment provided.
Response 1: As suggested by the reviewer, we revised the literature review part (by mentioning the specific point associated with each reference) in lines 69-99 as follows: There have been several studies concerning the influencing factors of water footprint. Bocchiola et al. (2013), Fulton et al. (2014), Ali et al. (2016), Miglietta et al. (2017), Mohammad et al. (2018), Nouri et al. (2019), and Mourad et al. (2019) have studied, respectively, the impact of climate change, policy change, human capital, gross national income, water harvesting technology, agricultural expansion and trade openness on a country's water footprint [1-7]. In the context of Chinese economy, scholars have found that population factors (Wang et al., 2014) [8], economic development levels (Zhao et al., 2014) [9], water conserving technology (Zhi et al., 2014) [10], international trade (Yang et al., 2015) [11], inward and outward foreign direct investment (Zhang et al., 2015; Kan and Huang, 2019) [12,13], climatic conditions (Yang et al., 2016) [14], consumption levels (Wang et al., 2019) [15], industrial structure (Xie et al., 2019) [16], water use efficiency (Kan and Lv, 2019) [17], geographical location (Zhang et al., 2019) [18], and shale gas development (Xu et al., 2019) [19] are important influencing factors of water footprint. The studies have not explicitly examined urbanization as an influencing factor of water footprint.
There have been several studies concerning the impact of urbanization on water quality. Most studies show that urbanization has a negative effect on water quality (Cerqueira et al., 2019; Freeman et al., 2019) [20,21]. Scholars have explored the impact of urbanization on water resources utilization in three aspects. The first aspect is the impact of urbanization on the amount of water resources utilization. Some studies found that urbanization led to the increase of total water use, and the impact was linear (Yang and Ding, 2014; Ma, 2014) [22,23]. However, some other studies found that the impact of urbanization on water resources utilization had a threshold effect, which was non-linear (Kan and Lv, 2017) [24]. In addition, urbanization’s impacts vary by different types of water consumption and by different levels of water consumption (Jin et al., 2018; Zhang et al., 2019) [25,26]. The second aspect is the impact of urbanization on the efficiency of water resources utilization. While some studies have found that urbanization improves water use efficiency (Bao and Chen, 2017; Wang, 2020) [27,28], Ding et al. (2019) have found that both population urbanization and land urbanization have a negative impact on industrial water utilization efficiency [29], some others have found that the relationship between urbanization and water use efficiency is inverted N-shaped (Cao, 2017) [30]. The third aspect is the impact of urbanization on the structure of water use. Most studies show that urbanization decreases the proportion of agricultural water use and increases the proportion of industrial water use and household water use (Lu et al., 2016; Cao, 2017) [31,32].
- Remove "billion" as it is often commuted with "milliard" due to discrepancies between the UK and US English usage, use mathematical symbols instead.
Response 2: In response to the reviewer’s comment, we used mathematical symbols instead of "billion" in lines 39, 131-148 and 166-167.
- Authors are using water footprint without proper referencing of the origin, definitions, and specification.
Response 3: As suggested by the reviewer, we have added the definition of water footprint in lines 64-65 as follows: water footprint refers to water resources needed in the production process.
- Please be more specific:
"Based on the existing literature and the generalized spatial panel specification, we construct the following model specifying industry water footprint (WF) as the dependent variable and urbanization (UR) as the focus independent variable:"
Response 4: As suggested by the reviewer, we have made revisions to make it more specific in lines 170-171 as follows: Based on the study of Kan and Lv (2018) and the generalized spatial panel specification (Lesage and Pace, 2009).
- In the conclusions, in addition to summarising the actions taken and results, please strengthen the explanation of their significance. It is recommended to use quantitative reasoning comparing with appropriate benchmarks, especially those stemming from previous work.
Response 5: As suggested by the reviewer, we have made revisions in lines 447-448, 453-456, 461-463, 483-486, 648-654 as follows: according to China Statistical Yearbook, China's reclaimed water reuse rate only accounts for about 11% of the sewage treatment capacity in 2015. The average quality of urbanization during the sample period is 1.697 in the eastern region; the average quality of urbanization during the sample period in the central and western regions is 1.112 and 0.975, respectively. According to China Statistical Yearbook, the reclaimed water reuse rate of the eastern region accounts for about 25% of the sewage treatment capacity in 2015. According to China Statistical Yearbook, the average agricultural production efficiency during the sample period is $920.9 per capita in the eastern region; the agricultural production efficiency during the sample period in the central and western regions is $731.0 per capita and $791.8 per capita, respectively. With the development of new urbanization, urbanization has changed from extensive mode to intensive mode. The world is facing many challenges from global climate change: global warming will directly affect the quantity of water resources, lead to reduction of glaciers and decrease water flow of rivers and lakes, and cause drought; it will also cause more water evaporation and decrease precipitation; it will also cause floods in coastal areas, and pollute fresh water resources. We hope the above proposed policy measures will help to reduce industry water footprint and to reconcile the contradiction between water supply and demand in the process of urbanization.
- The paper is focusing on China, however, it should not mean that the majority of references are from that region.
Response 6: As suggested by the reviewer, we have added 8 references that are not from China and deleted some references from China in lines 667-766.
- In English text, equations, and pictures, please use British standards for numbers in the text and pictures with delimiters:
1,000,000 rather than 1000000 or 1 000 000 or 1.000.000
Response 7: As suggested by the reviewer, we have used British standards for numbers in the text and pictures with delimiters in lines 50-56, 493, 501.
- For the text clarity would you refrain from using additional words, mostly meaningless filler words, which can be omitted or some archaic words see e.g. “respectively”, “thus”, “hence”, therefore”, “furthermore”, “thereby”, “basically,”, “meanwhile”,” wherein”, “herein”, “hitherto”, “Nonetheless”, “Perceivably”, “whereas”,etc. ?
Response 8: As suggested by the reviewer, we have made revisions in lines 346, 465, 493, 594.
- 210.38million RMB Should be 210.38 MRMB, however, you should also provide some indicative exchange rate between RMB and say EUR or USD statements as.
Response 9: As suggested by the reviewer, we have made revisions in lines 566-570 as follows: In 2015, the eastern region invested $1081.96´106 in wastewater treatment, which amounted to an average investment of $98.36´106 per province in the region. The central and western regions invested $346.35´106 and $472.89´106, respectively. The average investment per province in the central and western regions was $57.72´106 and $33.78´106, respectively, which were only 58.69% and 34.34% of the counterpart in the eastern region (data source: Compiled from the 2016 China Statistical Yearbook).
- "In 2015, China exported $70.68 billion and imported $116.88 billion, with a 432 deficit of $46.2 billion agricultural products." are presented without providing the reference for the source of the information.
Response 10: As suggested by the reviewer, we have provided the reference for the source of the information in lines 520-521, 526-527, 535-536, 571.
Round 2
Reviewer 1 Report
This version has been greatly improved. Therefore, I recommend the publication of this manuscript in Sustainability.
Author Response
1. This version has been greatly improved. Therefore, I recommend the publication of this manuscript in Sustainability.
Response 1: Thanks for your comments.
Reviewer 3 Report
Well made rebuttal.
Just a small issue, please make referencing as follows:
Bocchiola et al. (2013) [1], and now what they specifically did
Fulton et al. (2014) [2] and again what they specifically did
Ali et al. (2016), Miglietta et al. (2017), Mohammad et al. (2018), Nouri et al. (2019), and Mourad et al. (2019) have studied, respectively, the impact of climate change, policy change, human capital, gross national income, water harvesting technology, agricultural expansion and trade openness on a country's water footprint [1-7].
Also, you declared to remove fillers - hover unnecessary
respectively
was actually added by the author
When you clear the last formal points, the paper can be accepted.
Author Response
1. Well made rebuttal.Just a small issue, please make referencing as follows:
Bocchiola et al. (2013) [1], and now what they specifically did
Fulton et al. (2014) [2] and again what they specifically did
Ali et al. (2016), Miglietta et al. (2017), Mohammad et al. (2018), Nouri et al. (2019), and Mourad et al. (2019) have studied, respectively, the impact of climate change, policy change, human capital, gross national income, water harvesting technology, agricultural expansion and trade openness on a country's water footprint [1-7].
Also, you declared to remove fillers - hover unnecessary
respectively
was actually added by the author
When you clear the last formal points, the paper can be accepted.
Response 1: As suggested by the reviewer, we revised the literature review part (by mentioning the specific point associated with each reference) in lines 69-73 as follows: There have been several studies concerning the influencing factors of water footprint. Scholars have studied the impact of climate change (Bocchiola et al., 2013) [1], policy change (Fulton et al., 2014) [2], human capital (Ali et al., 2016) [3], gross national income (Miglietta et al., 2017) [4], water harvesting technology (Mohammad et al., 2018) [5], agricultural expansion (Nouri et al., 2019) [6] and trade openness (Mourad et al., 2019) [7] on a country's water footprint.